# Improving CAR T-Cell Persistence

**DOI:** 10.3390/ijms221910828

**Published:** 2021-10-07

**Authors:** Violena Pietrobon, Lauren Anne Todd, Anghsumala Goswami, Ofir Stefanson, Zhifen Yang, Francesco Marincola

**Affiliations:** 1Refuge Biotechnologies, Inc., Menlo Park, CA 94025, USA; angshu.goswami@gmail.com (A.G.); stefansonofir@gmail.com (O.S.); zhifen.yang@gmail.com (Z.Y.); 2Department of Biology, Faculty of Science, University of Waterloo, Waterloo, ON N2L 3G1, Canada; lauren.todd@uwaterloo.ca; 3Kite Pharma, Inc., Santa Monica, CA 90404, USA

**Keywords:** persistence, exhaustion, CAR, lymphodepletion, culturing conditions, stemness, TRUCK

## Abstract

Over the last decade remarkable progress has been made in enhancing the efficacy of CAR T therapies. However, the clinical benefits are still limited, especially in solid tumors. Even in hematological settings, patients that respond to CAR T therapies remain at risk of relapsing due to several factors including poor T-cell expansion and lack of long-term persistence after adoptive transfer. This issue is even more evident in solid tumors, as the tumor microenvironment negatively influences the survival, infiltration, and activity of T-cells. Limited persistence remains a significant hindrance to the development of effective CAR T therapies due to several determinants, which are encountered from the cell manufacturing step and onwards. CAR design and ex vivo manipulation, including culture conditions, may play a pivotal role. Moreover, previous chemotherapy and lymphodepleting treatments may play a relevant role. In this review, the main causes for decreased persistence of CAR T-cells in patients will be discussed, focusing on the molecular mechanisms underlying T-cell exhaustion. The approaches taken so far to overcome these limitations and to create exhaustion-resistant T-cells will be described. We will also examine the knowledge gained from several key clinical trials and highlight the molecular mechanisms determining T-cell stemness, as promoting stemness may represent an attractive approach to improve T-cell therapies.

## 1. Introduction

Adoptive cell therapy (ACT) is a type of immunotherapy in which patient-derived T-cells are genetically engineered ex vivo to target and kill cancer cells and are subsequently delivered back into the patient [1]. Such cells are modified to express synthetic receptors called Chimeric Antigen Receptors (CARs), which bind specific antigens on the surface of cancer cells and trigger a cytotoxic response [2]. First generation CARs were developed by Zelig Eshhar in 1993 and provided a T-cell receptor (TCR)-like ‘signal 1’ alone via CD3ζ or FcεRIγ [3]. However, limited persistence, expansion, and anti-tumor efficacy restricted their clinical benefits. Krause et al. (1998) demonstrated that by introducing the costimulatory domain of CD28 into the intracellular region of the CAR (second generation CARs), signals 1 and 2 could be combined to achieve increased efficacy, expansion, and persistence in vivo [4]. Third generation CARs combine two or more costimulatory domains from CD28, OX40, 4-1BB, or an inducible T-cell costimulatory (ICOS) domain. Despite their suggested increased performance in a handful of clinical studies, data analyzing the performance of third-generation CARs remain limited, and whether they have superior potency remains controversial [5,6,7,8,9,10].

Remarkable clinical success has been achieved using second generation anti-CD19 CAR T-cells to treat relapsed hematological malignancies in children and adults. CD19 is a B-lymphocyte surface antigen which is expressed in over 95% of B-cell malignancies including B-cell non-Hodgkin lymphoma (B-NHL), B-cell acute lymphoblastic leukemia (B-ALL), and chronic lymphocytic leukemia (CLL) [11,12]. Up to 60–70% of NHL patients and 90–95% of B-ALL patients have been found to respond to anti-CD19 CAR T-cell therapy, while the response rate is lower in CLL patients (50–70%) [12,13,14,15,16,17]. Considering these findings, the United States Food and Drug Administration has so far approved five CAR T-cell therapies, while the European Medicines Agency approved two. However, the overall survival for patients with relapses after CD19 CAR-T treatment is usually five months and further treatments options remain limited [18]. For example, Chow et al. (2020), suggested that treatments with an alternative CD19-targeting CAR-T product could improve patients’ survival [19]. In this study, axicabtagene ciloleucel (axi-cel) was administered to three patients exhibiting relapsed/refractory DLBCL after treatment with tisagenlecleucel or investigational CD19 CAR-T with 4-1BB costimulation. No positive result was observed [19]. Another example is the clinical trial #NCT01865617, where repeated infusions of the same CD19 CAR-T product were performed in patients with chronic lymphocytic leukemia, NHL, or acute lymphoblastic leukemia that have come back or have not responded to previous treatment. Higher cell doses in the second infusion and more intensive lymphodepletion were associated with better outcomes in some patients [20].

Despite the tremendous progress that has been made in enhancing the efficacy of CAR T therapies during the last decade, the clinical benefits are still limited, especially in solid tumors. Even in hematological settings, patients that respond to CAR T therapies remain at risk of relapse due to several factors including poor T-cell expansion and lack of long-term persistence after adoptive transfer [16,21,22]. This issue is even more dramatic in solid tumors, as the tumor microenvironment (TME) negatively influences the survival, infiltration, and activity of T-cells. Indeed, one of the hallmarks of cancer is the functional impairment of T-cells caused by a plethora of suppressive determinants in the TME, which leads to T-cell exhaustion [23,24,25,26,27]. Chronic antigen stimulation in the TME and immunosuppressive cytokines lead to dysfunctional T-cells, which are characterized by a peculiar continuum of epigenetic and transcriptional alterations resulting in the expression of inhibitory receptors and loss of early differentiation markers [28,29,30,31,32,33].

Limited persistence remains a significant hindrance to the development of effective CAR T therapies, thus engineering CAR T-cells to improve their survival and reverse or prevent the exhausted phenotype represents a logical therapeutic approach. Modern synthetic biology and genome-editing technology provide the opportunity to control exhaustion-specific pathways through the knock-out or knock-in of specific genes. Alternatively, epigenetic reprogramming (CRISPRi/a) is a powerful non-gene-editing synthetic biology methodology, which allows the upregulation or downregulation of target genes without permanently affecting DNA structure. In this review, the main determinants of decreased persistence of CAR T-cells in patients will be discussed, focusing on the molecular mechanisms underlying T-cell exhaustion. The approaches taken so far to overcome these limitations and to create exhaustion-resistant T-cells will be discussed, and several key clinical trials will be examined. We will also highlight the molecular mechanisms underlying T-cell stemness, as promoting stemness may represent an attractive approach to improve T-cell therapies by counteracting T-cell exhaustion.

## 2. Factors Influencing T-Cell Persistence in Clinical Settings

Several factors impact the persistence of CAR T-cells in patients, including activation-induced cell death (AICD) and peripheral tolerance, which may be described as a state of antigen recognition but lack of reactivity toward cancer cells [34]. Tolerance occurs in situations of chronic low-grade inflammation (para-inflammation) and it is dependent on the context of different cancer subtypes. Indeed, it has been demonstrated that a modest increase in cytokine levels may be linked to immune tolerance [35]. Cytokines such as IL-6, IL-1 and IFN-γ are involved in the development of immune tolerance and, considering that different tumors may present a different cytokine profile in the TME, tolerance mechanisms may also vary in cancer subtypes [36]. Peripheral ignorance, which is due to the lack of tissue inflammation, is responsible for the absence of T cell recruitment instead [37]. Moreover, immunological clearance due to CAR rejection by the host immune system has been reported in several clinical trials. Therefore, the molecular design of the CAR construct and the ex vivo culturing procedures play important roles in shaping the response to CAR T-cell therapy, in terms of both potency and duration [38,39,40,41]. In this section, we will discuss the determinants of CAR T-cell therapy efficacy and highlight that peak expansion and persistence should be considered when choosing a CAR T-cell product.

### 2.1. CAR Construct

The type of extracellular domain (single chain variable fragment scFv), the type of spacer domain which links the scFv to the transmembrane domain, and the costimulatory domain that composes the CAR, have been demonstrated to profoundly impact T-cell function and persistence (Figure 1) [39,42,43,44,45]. Most CAR T-cells used in clinical trials are derived from mice, which could lead to both humoral and CD8^+^ T-cell-mediated immune responses resulting in immunological rejection. The reinfusion of CAR T-cells containing murine components into patients that experienced CD19^+^ relapse after the first treatment is largely ineffectual [22,46,47,48,49,50,51]. Furthermore, murine clusters of CAR receptors on the cell surface could result in tonic signaling, leading to T-cell exhaustion [52]. CARs with humanized single chain variable fragment (scFv) regions entered clinical trials in 2016, when the first 100% human-derived anti-CD19 CAR (HuCAR-19) was administered to nine patients with advanced NHL [53]. This trial concluded that HuCAR-19 CAR T-cells have substantial activity against advanced lymphoma, with a reported overall response rate (ORR) of 86%. Humanized CAR T-cells have shown similar cytotoxic activity as murine CAR T-cells and enhanced persistence due to lower immunogenicity [51]. They also displayed decreased T-cell exhaustion due to reduced antigen-independent tonic signaling [54,55,56]. However, the creation of 100% human-derived CARs remains technically challenging.

CAR binding affinity affects tonic signaling and excessive affinity for the target may lead to T-cell exhaustion. Ghorashian et al. (2019), developed a CAT CAR (CD19 scFv) with 40-fold lower affinity to CD19 than the conventional CAR derived from FMC63 [57]. This construct was tested in patients with relapsed or refractory pediatric BLL, and the clinical study (CARPALL, #NCT02443831) found that 12 of 14 patients achieved molecular remission without severe cytokine release syndrome (CRS). Persistence was demonstrated in 11 of 14 patients with increased CAR T-cell expansion.

The costimulatory domain also plays a crucial role in CAR T-cell persistence and efficacy. Examples of costimulatory molecules are CD28, ICOS, CD27, 4-1BB, OX40, and CD40L. Several studies found enhanced T-cell persistence in CAR constructs equipped with a 4-1BB domain rather than a CD28 domain [58,59,60]. However, other studies reported conflicting evidence in the difference in persistence between the two costimulatory domains, probably due to variability among patients and tumor types [6,61,62,63,64,65]. Other studies have reported that CAR T-cells containing 4-1BB display increased memory T-cell (TEM) phenotypic markers, which may delay CAR T-cell exhaustion [6,52,66]. Insights into the molecular pathways downstream of CAR T-cell activation may explain the observed differences in the persistence of CAR T-cells harboring CD28 or 4-1BB. Indeed, it has been demonstrated that CD19-28ζ T-cells exhibit increased activation of the transcription factor NF-κB, which is responsible for the induction of cytokine gene transcription and secretion [67,68,69,70]. Therefore, these cells are characterized by high cytotoxic efficacy and expansion, but also rapid exhaustion due to tonic CAR CD3ζ phosphorylation caused by the clustering of CAR scFv [6,52]. They are also responsible for severe CRS, and patients infused with CD19-28ζ T-cells may need steroid therapy, which in turn limits CAR T-cell persistence. In contrast, CD19- BBζ T-cells showed better persistence and mitigated exhaustion and apoptosis [22,71,72,73]. The 4-1BB domain has been shown to be responsible for the recruitment of TRAF proteins, activation of the AKT/mammalian target of rapamycin (mTOR) pathway, and upregulation of the antiapoptotic genes Bcl-xL and BFL-1 [74,75,76,77,78,79]. Recent evidence suggests that in BBζ CAR T-cells, recruitment of the SHP-1 phosphatase leads to a reduction in CAR signaling [80]. In 28ζ CAR T-cells, recruitment of the SHP-1 CAR T-cells, constitutive phosphorylation of CD3ζ causes the accumulation of Nuclear Factor of Activated T-cells (NFAT) homodimers that are responsible for the exhausted phenotype [81]. The inclusion of 4-1BB in the CAR also promotes the expansion of stem memory T-cells (TSCM), respiratory capacity, mitochondrial biogenesis, and fatty acid oxidation (FAO). CD28 is associated with a very different metabolic profile, enhancing glycolysis, and favoring the expansion of effector TEMs [66,82,83,84,85,86,87].

ICOS belongs to the CD28 family of costimulatory molecules and has been shown to be essential for appropriate immune responses. Guedan et al. (2018) demonstrated that the combination of both ICOS and 4-1BB costimulatory domains in the CAR construct significantly increase T-cell persistence, however ICOS is beneficial only when positioned proximal to the cell membrane and linked to the ICOS transmembrane domain [88]. Guedan et al. (2020) also demonstrated that mutating an asparagine to a phenylalanine (YMFM) in the costimulatory domain of CD28-based anti-mesothelin CAR T-cells improved persistence and functionality in a xenograft model of pancreatic cancer [81]. This type of CAR T-cell still demonstrates shorter persistence compared to those containing ICOS or 4-1BB due to increased NFAT signaling, which drives T-cell exhaustion. Mice treated with ICOS-CAR T-cells and CD28-YMFM CAR T-cells showed similar tumor control and T-cell persistence. Interestingly, the reciprocal substitution in the ICOS domain (phenylalanine to asparagine) decreased in vivo functionality [81]. Riddell’s group introduced a mutation in the Lck-binding domain of CD28 that increased its efficacy by suppressing CAR signaling, which mitigated the exhausted phenotype [59]. Sadelain’s group developed innovative TRAC-CAR T-cells, in which CAR cDNA is integrated through CRISPR-Cas9 into the TRAC locus [89]. This strategy permits the downregulation of surface CAR expression upon antigen exposure, resulting in decreased exhaustion in vivo. These observations suggest that the configuration of the costimulatory domain is an important consideration in the generation of CAR constructs.

Despite conflicting evidence, it is recognized that CD28-CAR is associated with high effector function and limited T-cell persistence, while 4-1BB-CAR and ICOS-CAR are less potent but last longer. Therefore, calibration of CAR activation potential is now considered an important requirement for the success of clinical trials. Feutcht et al. (2019) hypothesized that the redundancy of CD28 and CD3ζ signaling in a CAR incorporating all three CD3ζ immunoreceptor tyrosine-based activation motifs (ITAMs) may enhance T cell exhaustion [90]. They performed an ITAM dosage, calibrating ITAM activity by mutating tyrosine residues to impede their phosphorylation and downstream signaling. This study showed that CARs encoding a single ITAM resulted in T-cells with balanced effector and memory programs and enhanced therapeutic profiles. Also, in the past few years novel CAR T products have been investigated, which combine the CD28 and 4-1BB costimulatory domains [91]. Indeed, Drent and colleagues showed that CAR-T cells harboring both CD28 and 4-1BB costimulatory domains present improved in vivo antitumor function, increased proliferation and maintenance of a central memory phenotype, compared to CAR-T cells equipped only with the 4-1BB domain.

### 2.2. Ex Vivo Manipulation and Patient Lymphodepletion

The process of T-cell manufacturing requires the harvesting of an adequate number of healthy T-cells from patients. However, previous chemotherapeutic treatments, especially those containing clofarabine or doxorubicin, result in lymphopenia or suboptimal quality of the final CAR T-cell products (Figure 1). Cytotoxic treatments affect global metabolic pathways in patient’s T-cells, leading to poor-quality T-cells. The type of T-cells used for infusion critically impact the success of therapy. Indeed, better engraftment in preclinical models has been obtained by normalizing CD4/CD8 ratios or using naïve or memory cell subsets [46,92]. Several studies have demonstrated that T-cell populations enriched for early lineage cells expand better and lead to improved persistence and efficacy of the CAR T-cell treatment [93,94]. Singh et al. (2016) investigated T-cell expansion and memory phenotypes during chemotherapy in pediatric patients with ALL and NHL and found that cyclophosphamide and cytarabine depleted early lineage cells. However, culturing the harvested cells with IL-7 and IL-15 rescued expansion capabilities and enriched for select early lineage cells [94]. Other clinical trials (#NCT00586391 and #NCT00709033) and in vitro experiments have supported this observation. For example, IL-7 and IL-15 can induce naïve T-cells to expand into effector memory T subsets, preserve the stem memory T cell (TSCM) phenotype, and increase resistance to apoptosis following repetitive stimulation, while preserving the migration to secondary lymphoid organs [95,96,97,98,99,100]. Multiple studies showed that the addition of antioxidants such as N-acetylcysteine to the cell culture during the manufacturing process has also been shown to inhibit effector differentiation and promote the expansion of TSCM cells [101,102,103].

A few days before CAR T-cell infusion, a lymphodepleting chemotherapy regimen is administered to patients, the most common being cyclophosphamide (cy), fludarabine (flu), and bendamustine (ben). Lymphodepletion treatments eradicate regulatory T-cells (Tregs) and other immunosuppressive cells increasing CAR T-cell expansion and prolonging their persistence [104,105]. Adequate lymphodepletion is crucial for therapeutic success and may prevent CAR rejection, as observed in adult B-ALL and B-NHL clinical trials [106,107].

In a clinical trial, 12 adults treated with only cy chemotherapy experienced limited CAR T-cell persistence, similar to what has been reported by other trials in which rejection of the murine CAR components was observed [107]. Subsequently, flu was added to the lymphodepleting regimen of 16 patients. This led to improved results such that an ORR of 67% and complete remission were observed in several patients [106]. Moreover, a recent study on patients with Hodgkin lymphoma undergoing CD30 CAR T-cell therapy showed that combining ben and flu results in longer CAR T-cell persistence compared to a combination of cy and flu [108]. Compared to ben alone, the ben-flu combination increased the levels of circulating IL-15 and IL-17 and significantly increased progression-free survival in patients, compared to ben alone or a combination of cy and flu.

Interestingly, Hirayama et al. (2019) performed an extensive multivariate analysis of factors to identify markers associated with a complete response and progression-free survival (PFS) [20]. The clinical trials #NCT01865617 showed that patients with aggressive B-NHL receiving high intensity cy-flu lymphodepletion were more likely to exhibit a favorable cytokine profile (MCP-1 and IL-7), compared to patients with low intensity treatment. Among patients treated with high dose lymphodepletion, PFS was increased in those achieving a favorable cytokine profile as well, compared to patients that did not achieve a favorable profile [20]. Therefore, it seems that the biological effects derived from lymphodepleting treatments are likely more relevant than the intensity of the treatment itself to improve the efficacy of CAR T-cell therapy.

### 2.3. T-Cell Exhaustion

The adoptively-transferred CAR T-cells that reach and home into solid tumors after infusion encounter an immunosuppressive environment characterized by physical, functional, and dynamic barriers hindering T-cell function (Figure 1). Such barriers have been extensively discussed in comprehensive reviews [26,27,37,109]. In the TME, the presence of immunosuppressive cytokines produced by myeloid-derived suppressor cells (MDSCs), cancer-associated fibroblasts (CAFs), and tumor cells contribute to T-cell exhaustion. For example, the TGF-β signaling cascade is a main player during T-cell exhaustion, and it has been shown that knocking out endogenous Tgfbr2 in CAR T-cells prevents exhaustion by blocking TGF-β-induced phosphorylation of SMAD proteins. Knock out of the TGF-β receptor leads to increased expansion and cytotoxic potential in vitro and in vivo [110].

T-cell exhaustion can also be caused by chronic exposure to an antigen, and despite initial effective anticancer responses, CAR T-cells can progressively languish. During the interaction between T-cells and tumor cells, integrated signals coming both from cell-to-cell contacts and from cytokine-dependent crosstalk regulate T-cell expansion, differentiation, and persistence [29,30,111]. Two main pathways involved in this transduction network are the mitogen-activated protein kinase (MAPK) and PI3K/AKT/mTOR pathways [112,113,114]. MAPKs, a family of serine/threonine kinases, regulate a variety of cellular responses to stimuli including proliferation, migration, and differentiation. The PI3K/AKT/mTOR pathway cross talks with MYC, which regulate glucose metabolism and metabolic transcriptional profiles, respectively. Indeed, effector T-cells rely mainly on glycolytic metabolism to perform their cytotoxic functions, while the metabolism of their precursors consists of FAO [115].

If the antigen stimulation persists, T-cells become chronically exposed and enter a continuum of epigenetic, metabolic, and transcriptional alterations characterizing the state of exhaustion [116]. The PI3K/AKT pathway may also play a role in the progression of CAR T-cell exhaustion, as it was shown to reduce in vivo persistence of CD33-specific CAR T-cells in an acute myeloid leukemia model [113]. Treatment with a PI3K inhibitor improved in vivo persistence and efficacy and preserved a less differentiated state. Knockout of Ptpn2 in CD8^+^ T-cells increased the formation of terminally exhausted T-cells compared to progenitor exhausted T-cells [117]. Recent evidence suggests that the PTPN2 and PP2A phosphatases play a critical role in CD8^+^ T-cell exhaustion, as PTPN2 attenuates the type I interferon pathway while PP2A mediates CTLA-4 signaling [118,119].

As T-cell stimulation continues, T cells present a progressive loss of effector functions and reduced cytokine production [120,121]. Such a process occurs in a hierarchical manner, with IL-2 and TNF-α being lost early on. IFN-γ and chemokine secretion decrease later in the exhaustion progression. Despite high proliferative capacity being lost early as well, exhausted T-cells are still capable of limited proliferation when stimulated in vivo. During the exhaustion progression, the chronic TCR stimulation leads to modifications in T-cell receptor (TCR)-dependent signaling pathways, such as those mediated by nuclear factor of activated T-cells (NFAT) [122,123,124].

CD8 T-cells generate energy through glycolysis, oxidative phosphorylation (OXPHOS) and FAO [125]. These processes metabolize glucose and fatty acids to generate energy by producing intermediates that enter the tricarboxylic acid (TCA) cycle. TCA cycle produces electron carriers required in adenosine triphosphate (ATP) synthesis, which occurs through the electron transport chain. In addition to these pathways, CD8 T-cells also utilize alternate inputs to the TCA through the metabolism of glutamine and glutamate [126,127].

Naïve T cells rely primarily on OXPHOS and FAO to produce intermediates that enter the TCA cycle and lead to the formation of ATP molecules. TCR-based activation results in a shift of their metabolism from catabolic (OXPHOS) to anabolic (glycolysis) and in the upregulation of glutaminolysis and the pentose phosphate pathway [128,129,130,131,132]. While the precise regulation of the exhausted metabolic state remains unclear, chronic antigen stimulation leads to increased dependence on FAO and OXPHOS, which creates bioenergetic deficiencies, dampening CD8 effector functions even further.

Exhausted cells also present high expression of inhibitory receptors such as Programmed Cell Death Protein 1 (PD-1), T-cell immunoglobulin and mucin-domain containing-3 (TIM-3), Lymphocyte-activation gene 3 (LAG-3), CD160, B- and T-lymphocyte attenuator (BTLA), cytotoxic T-lymphocyte–associated antigen 4 (CTLA-4), and T cell immunoreceptor with Ig and ITIM domains (TIGIT) (Figure 2) [133,134,135,136,137,138]. Programmed Cell Death Ligand 1 (PD-L1) is a transmembrane protein expressed on the surface of certain cancer cells and is a co-inhibitory factor of the immune response. PD-L1 can bind PD-1, which is expressed on the T-cell surface. PD-1 promotes self-tolerance, modulates the activity of T-cells, activates apoptosis of T effector cells and inhibits apoptosis in regulatory T-cells (Tregs) [139,140,141,142]. When PD-L1 binds to PD-1, there is a reduction of the host immune response against cancer with decreased proliferation, cytokine secretion, and persistence of T-cells. However, the presence of inhibitory receptors is not the main feature of exhausted T-cells, as PD-1 is induced during the cytotoxic response in functional effector T-cells [143].

In the TME, the exhausted T-cell population is heterogeneous and is composed of T-cells at different stages of the exhaustion process, but also progenitors of exhausted T-cells [144,145]. For example, the PD1^+^ TCF1^+^ CD8^+^ T-cell subset demonstrates a link between T-cell memory and exhausted cells. These cells display a simultaneously exhausted phenotype and stem cell-like properties, and have been observed in both human and murine cancers. This cell subset can expand to give rise to PD1^+^ TCF1^−^ T-cells while maintaining the pool of progenitor cells. Another example is the PD1^+^ TCF1^+^ CXCR5^+^ TIM3^−^ progenitor stem cell-like population, which produces PD1^+^ TCF1^−^ CXCR5^−^ Tim3^+^ terminally differentiated exhausted cells [146,147,148,149]. Evidence suggests that exhausted CD8^+^ tumor infiltrating lymphocytes (TILs) include a progenitor subpopulation with self-renewal capability that can differentiate into exhausted T-cells in an antigen-dependent process. The process is initiated from TCF1^+^ exhausted T progenitors, followed by an intermediate state and a final state in which cells are terminally exhausted. In fact, PD1^+^ TCF1^+^ cells were detected among TILs and tumor reactive CD8^+^ T-cells, in melanoma patients [150,151]. Progenitor exhausted CD8^+^ TILs were able to respond to checkpoint blockade therapy and better control tumor growth, in contrast to terminally exhausted TILs. PD1^+^ TCF1^+^ TILs differentiated in Tcf1^+^ PD-1^+^ and Tcf1^−^ PD-1^+^ cells, and removal of Tcf1^+^PD-1^+^ population limited the response to immunotherapy. Indeed, patients with a higher percentage of progenitor exhausted cells experienced a longer duration of response to checkpoint blockade therapy [151].

An interesting model for T-cell exhaustion was suggested by Wherry and Kurachi (2015), according to which a variety of transcription factors regulate a hierarchical developmental pathway leading to the terminally exhausted phenotype [29]. Therefore, the epigenetic and transcriptional landscape displayed by exhausted T-cells is a continuum of different states and is closely associated with their dysfunctional state, and progression is finely regulated by a multitude of determinants [152,153].

In exhausted cells, epigenetic regulators including DNA methyltransferases (DNMT1 and DNMT3B) and histone modification proteins are significantly upregulated [154]. DNA methyltransferase 3A (DNMT3A) has been shown to establish a *de novo* exhaustion-specific DNA methylation pattern [155]. The expression of factors responsible for memory cell maintenance such as TCF1 and CCR7 are downregulated. Therefore, DNMT3A inhibits T-cell lineage plasticity [156]. DNMT3A drives T-cell exhaustion, like Ten Eleven Translocation (TET2) but by acting at distinct loci [157]. TET2 is a family of methylcytosine dioxygenases that catalyze DNA methylation. Several studies have shown that the inactivation of TET2 promotes anti-tumor immunity by inducing epigenetic modifications that reverse the transcriptional profile in cells undergoing exhaustion [158,159,160].

The NFAT family of transcription factors consists of five members, but T-cells express only calcium-regulated NFAT1, NFAT2, and NFAT4. Following T-cell activation, the levels of intracellular calcium are increased which activates calcineurin, causing nuclear translocation of NFAT. NFAT binds to the transcription factor activator protein 1 (AP1) and activates genes involved in T-cell activation and upregulation of effector cytokines (IL-2 and IL-4) [161]. However, in the absence of AP1, NFAT binds to different promoters and triggers modifications in the expression of genes that are required for anergy and exhaustion [162,163]. Some of the transcription factors downstream of NFAT include thymocyte selection-associated high mobility group box protein (TOX), Nuclear Receptor subfamily 4 group A member (NR4A) 2, and NR4A3, Eomes, IRF4, and HELIOS [164,165,166]. Downregulation of TOX mitigates the exhausted phenotype through increased effector function in vivo [167]. However, other studies have shown that depletion of TOX and TOX2 in tumor-specific T-cells leads to decreased persistence, suggesting that the biological relevance of TOX may extend beyond merely promoting the exhausted phenotype [168]. Another transcription factor regulated by NFAT is NR4A, which synergizes with TOX to induce T-cell exhaustion. In vivo, T-cells depleted for NR4A demonstrated increased cytokine production and surface markers similar to effector T-cells [167,169,170]. Chen et al. (2019) showed that NR4A transcription factors play a crucial role in the cell-intrinsic program of hypo-responsiveness [169]. Indeed, the Nr4a triple-knockout (Nr4a1, Nr4a2 and Nr4a3) in CAR TILs promoted tumor regression and prolonged the survival of tumor-bearing mice. Moreover, the gene expression profile in Nr4a triple knockout TILs showed unique features compared to wild type CD8, with enrichment in binding motifs for AP1 and NF-κB transcription factor [169]. Evidence showed that NR4A1 is recruited to AP1 binding sites, which reduces AP1 function and inhibits effector gene expression. The major epigenetic modification associated with T-cell exhaustion is indeed the dysregulation of the AP1-binding motifs, which leads to an increase in the expression of specific transcription factors [171]. Overexpression of the c-Jun oncogenic transcription factor directly activates the AP1 complex (c-Jun+c-Fos) and indirectly disrupts the inhibitory APi complex, which are associated with the exhaustion process. The overexpression of c-Jun rescued CAR T-cell exhaustion in vivo, increasing their functional capacities and reducing exhaustion markers [171].

The Eomes and T-bet transcription factors are crucial during the process of T-cell exhaustion. PD-1 blockade decreases Eomes expression, suggesting a dominant role for the two in terminally exhausted CAR T-cells [152]. Although Eomes knockdown partially rescued the exhausted phenotype, evidence showed that it is important for the maintenance of central memory T-cells (TCMs) [172,173,174]. In addition, Li et al. (2018) showed that complete loss of Eomes impaired the development of anti-tumor cytotoxic T lymphocytes (CTLs), whereas deletion of one allele mitigated CD8^+^ T-cell exhaustion and offered better tumor control [172]. It is plausible that the subcellular localization of Eomes and T-bet is crucial for their regulatory function during the exhaustion progression. Following acute infection, T-bet promotes terminal differentiation and represses inhibitory receptors during chronic viral infection. During chronic infection and in human tumors, T-bet is downregulated and it has been demonstrated that exhausted T-cells present a higher ratio of nuclear Eomes/T-bet, compared to memory T-cells [175,176]. High nuclear levels of T-bet repress *Pdcd1* transcripts, while low nuclear levels in exhausted cells lead to a dominant effect of Eomes, which is a weaker repressor of *Pdcd1.* In fact, T-bet and Eomes compete for the same binding motifs, including the *Pdcd1* T-box (PD-1 gene) [175]. Not only T-bet directly represses *Pdcd1*, but it was also shown to decrease other inhibitory receptors. McLane et al. (2021) have also demonstrated that blocking PD-1 signaling in exhausted cells increases nuclear T-bet and restores full repression of *Pdcd1.* In Crohn’s disease, T-bet may be responsible for the development and persistence of the inflammation as its overexpression seems to interfere with PD-1/PDL-1 mediated suppression of CD4^+^ T cell survival [177]. However, the interaction between Eomes and T-bet is still not fully characterized and warrants further investigation.

In conclusion, a plethora of transcription factors has been found to be involved during the progression of T-cell exhaustion. [178,179,180,181,182,183,184,185,186,187,188]. Therefore, the regulation of the progression into exhaustion is due to multifaceted interactions and a combination of different epigenetic and transcriptional factors.

In this chapter, we discussed the exhaustion phenomenon, which is a major cause of the lack of CAR T-cell persistence in tumors. We described some of the most important mechanisms and determinants involved in T-cell exhaustion, and in the following chapter, we will highlight recent attempts to overcome this issue.

## 3. Strategies to Improve Car T-Cell Persistence and Potency

### 3.1. Car-T-Cells Engineered to Express Cytokine and Their Receptors

A fourth generation of CAR T-cells has recently been developed to resist the immune-hostile environment in the TME while simultaneously overcoming immune exhaustion. “T-cells redirected for antigen-unrestricted cytokine-initiated killing” (TRUCKs) have been designed to combine the cytotoxic function of CAR T-cells with the in situ delivery of cytokines, which have immune modulating capacities. CAR T-cells are equipped with a constitutive or inducible expression cassette coding for a cytokine or any other transgenic protein of interest. In the case of an inducible system, after the CAR binds the antigen, the cytokine is synthesized and can act in an autocrine fashion to increase the survival and amplification of the T-cell. Cytokines can also act in a paracrine fashion, modulating the surrounding environment and interfering with the immunosuppressive cytokine profile present in the TME. A panel of cytokines including IL-12, IL-7, IL-15, IL-18, IL-21 and IL-23 are currently being investigated and entering early phase clinical trials [189,190,191,192].

IL-12 is a pro-inflammatory cytokine that induces a Th1 CD4^+^ T-cell response to promote CD8^+^ clonal expansion and persistence. It is also responsible for the regulation of cytotoxic activity in CTLs and natural killer (NK) cells, reactivating anergic tumor-infiltrating lymphocytes, recruiting NK cells, and inhibiting Tregs. IL-12 also inhibits the secretion of immunosuppressive cytokines by tumor-associated macrophages (TAMs) including suppression of TGF-β secretion [193,194,195]. Preclinical studies in CD19-targeting CARs constitutively expressing IL-12 demonstrated that it enhances tumor killing efficacy and immune memory against the cancer antigen [196]. However, the potentially lethal toxicity associated with constitutive secretion of IL-12 renders necessary the development of an inducible system limiting IL-12 secretion exclusively upon CAR activation [197]. A clinical trial (NCT01236573) examined the infusion of CD8^+^ T-cells conditionally expressing IL-12 driven by NFAT induction upon T-cell activation. Low persistence of IL-12-transduced TILs was observed, and the study was terminated due to severe toxicity. NCT02498912 is an active clinical trial investigating the functionality of autologous T-cells engineered to secrete IL-12 and target the MUC16ecto antigen in patients with solid tumors. NCT03932565 is currently in the recruitment stage and will study fourth generation Nectin4-targeting CAR T-cells secreting IL7 and CCL19 or/and IL12 in solid tumors. Another planned clinical trial (NCT03542799) will evaluate EGFR-CAR T-cells engineered to secrete IL-12 upon CAR activation via NFAT induction in patients with metastatic colorectal cancer.

IL-15 is a cytokine that stimulates CD8^+^ T-cell and NK cell activation, proliferation, and cytotoxic activity [198]. IL-18 augments Th1 cell cytokine production while inhibiting the synthesis of IL-10 [199,200]. Both IL-15 and IL-18 are enhancers of the immune response and have been tested in TRUCKs. IL-18- and IL-15-secreting TRUCKs showed enhanced expansion and persistence in mice bearing tumors, compared to conventional CAR T-cells, and showed increased tumor cytotoxicity in vitro and in vivo [201,202,203,204]. NCT04684563 is a clinical trial recruiting patients with NHL and CLL that aims to find the maximum dose of huCAR19-IL18 cells that is safe for use. Patients will be infused with anti-CD19 CAR T-cells which also express IL-18. Six clinical trials are currently recruiting to test IL-15-secreting TRUCKs in solid and hematological malignancies using engineered T-cells and NK cells.

IL-7 is required at all stages of T-cell development and is also involved in regulating T-cell survival and mature T-cell homeostasis [205,206]. In 2006, IL-7 was tested in clinical trials to increase the population of CD8^+^ (CD45RA^+^ CCR7^+^ and/or CD27^+^), CD4^+^ (CD45RA^+^ CD31^+^), and CD4^+^ central TEMs (TCM) (CD45RA^−^CCR7^+^). Remarkably, IL-7 treatment also broadened patients’ naïve T-cells repertoire compared to before the treatment [207].

Recent studies demonstrated that it is possible to co-express multiple cytokines in the same CAR T-cell. For example, Luo et al. (2020) engineered CAR T-cells to produce IL-7 and CCL21 and demonstrated that this combination improved survival and infiltration of CAR T-cells in solid tumors in vivo, without requiring preconditioned lymphodepletion [208]. NCT04833504 is a clinical trial recently completed in which CD19^+^ CAR T-cells expressing IL-7 and CCL19 were tested in patients with relapsed or refractory B-cell lymphoma, however the results have not yet been reported. Two additional clinical trials are currently recruiting to test TRUCK cells expressing IL-7 in combination with a PD1 blockade or with the secretion of other cytokines. Mohammed et al. (2017) developed a switch receptor in which the extracellular domain binds IL-4, a negative regulator of T-cell function which is present in the TME [209]. The intracellular domain instead consists of the endodomain of the IL-7 receptor. This “switch” construct turns the IL-4 inhibitory signal into an inducer of T-cell fitness. The same was carried out for IL-4/IL-21, and an improvement in the efficacy of CAR T-cell therapy was reported [210]. Indeed, TRUCK cells can also be engineered to resist immunosuppressive factors in the TME, such as TGF-β-mediated inhibitory signals. In many cancers, the secretion of TGF- β constitutes a functional barrier that inhibits the immune response and favors cancer progression. Blocking the TGF-β signaling cascade in T-cells increases their ability to infiltrate, proliferate and mediate cytotoxic responses. This has been demonstrated by Kloss et al. (2018) in a prostate cancer model using CAR T-cells harboring a truncated form of TGF-β receptor (dnTGF-βRII), which lacks the intracellular domain, pivotal for the downstream signaling propagation [211]. Anti-PSMA CAR T-cells co-expressing the dnTGF-βRII displayed increased proliferation, cytokine secretion, resistance to exhaustion and persistence, in mouse models. In the light of these results, the single arm Phase I clinical trial NCT03089203 is currently active (but not recruiting) to test the efficacy of such CAR T-cells in patients with metastatic castrate resistant prostate cancer.

### 3.2. Checkpoint Blockade Therapy

Immune checkpoints regulate self-tolerance by preventing the immune system from attacking cells indiscriminately. Important checkpoint proteins are PD-1, LAG3, TIM3, TIGIT, CTLA-4, GITR, and BTLA [212,213,214]. PD-L1 is present on the surface of tumor cells, TAMs and MDSCs. When PD-L1 engages with PD-1 on T-cells, it triggers T-cell apoptosis and the depletion of TILs in the TME. Moreover, the RAS/MEK/ERK and PI3K/AKT pathways are inhibited, which blocks cell proliferation. Finally, activation of PD-1 also favors the transformation of CD4^+^ T-cells into Tregs [141,215]. PD-1 is the most common inhibitory receptor expressed on the surface of exhausted CD8^+^ T-cells. Its dynamics are particularly interesting because its expression is induced by T cell activation and it can be, in that sense, seen as an activation marker though its functions remain consistently suppressive [216,217]. In activated T-cells, CTLA-4 competes for CD28 binding and initiates immunosuppressive signals such as PD-1, inducing the PI3K/AKT pathway [218,219]. CTLA-4 downregulates TCR activation upon binding to the B7 protein [220,221]. Interestingly, in Tregs, CTLA-4 is constitutively expressed and provides activation signals [222]. The TIGIT immune checkpoint binds the ligands CD155 and CD112 and is transiently overexpressed in exhausted CD8^+^ T-cells, Tregs, and NK cells [223,224,225,226]. The molecular mechanisms and signaling cascades underlying TIGIT, TIM-3, and LAG-3 activation and exhaustion induction remain less understood.

Strategies to decrease exhaustion by repressing checkpoint signaling include knocking down or knocking out co-inhibitory molecules through shRNA-expressing vectors or CRISPR/Cas9. The use of co-inhibitory molecules such as anti-PD1/PDL-1 monoclonal antibodies and inhibition of specific transcription factors involved in the expression of checkpoints have also been tested. Many studies reported that blocking checkpoint suppression restores cytokine production and promotes CAR T-cell survival [227,228,229,230]. Moreover, the concurrent blockage of multiple immune checkpoints such as PD-1, TIM-3, and LAG-3 synergistically increases the effector functions of CAR T-cells [231,232,233].

Combining CAR T-cells with treatments involving immune checkpoint blockades may represent a useful strategy to enhance antitumor activity, persistence, and memory cell formation. Immune checkpoint blockades combining anti-PD-1 and/or anti-CTLA-4 antibodies have been shown to improve the treatment of solid and hematological tumors [234]. In glioblastoma and breast cancer cell lines, the administration of anti-PD1 antibodies enhanced the antitumor activity of anti-HER2 CAR T-cells [235,236]. However, some clinical trials reported negative results in neuroblastoma patients after combined treatment with PD-1 inhibitors and anti-GD2 CAR T-cells [237]. The anti-PD-1 drugs currently used in clinical settings are Pembrolizumab (Keytruda), Nivolumab (Opdivo), and Cemiplimab (Libtayo). Anti-PD-L1 inhibitors have also been used in patients, and the currently approved drugs include tezolizumab (Tecentriq), Avelumab (Bavencio), and Durvalumab (Imfinzi). Additionally, Ipilimumab (Yervoy) and tremelimumab are two immune checkpoint inhibitor drugs that block CTLA-4.

Other strategies have been pursued to block PD-1, such as genome editing to equip CAR T-cells to secrete a PD-1 blocking antibody or downregulating PD-1 [229,238,239,240]. This technique makes it possible to render CAR T-cells unresponsive to the PD-1/ PD-L1 suppressive signals in the TME. Checkpoint-resistant CAR T-cells have also been created using a one-shot CRISPR system, by knocking out TCR, HLA class I, PD-1, and CTLA-4 [89,230,241,242]. Alternatively, engineered cells harboring the extracellular portion of the PD-1 domain fused to the intracellular stimulatory domain of CD28 were tested [243,244]. Despite that PD-1 is an exhaustion marker, Odorizzi et al. (2015) reported that knock out of *Pdcd1* may itself promote exhaustion and impair T-cell survival and function [245]. This could be explained by the fact that PD-1 expression is a marker of T-cell activation and is transiently expressed in naïve T-cells upon TCR activation. For example, in melanoma tumors, PD-1 facilitated the identification of tumor-reactive CD8^+^ T-cells, and the level of PD-1 expression was associated with the strength of TCR signaling [246]. However, PD-1 expression shifts from transient to constitutive when the antigen persists, and T-cells become chronically activated. Such phenomena underlie the complexity of the biological significance of PD-1 to T-cells.

The exact molecular mechanisms underlying the efficacy of immune checkpoint blockades are still unclear. However, checkpoint blockade does not alter the epigenetic state of exhausted cells, and thus does not reverse exhaustion [32,151,152,155,247]. Moreover, it was recently demonstrated that not all PD-1^+^ cells respond equally to anti-PD-1 treatment. For example, PD-1^+^ CD38^+^ CD8^+^ T-cells are a population of dysfunctional cells that fail to respond to anti-PD-1 therapy [248]. Interestingly, evidence suggests that a population of exhausted TILs expressing TCF1 respond to anti-PD1 treatment and can transform into self-renewing memory cells [150]. Furthermore, the PD1^+^ TCF1^+^ CD8^+^ T-cell subset suggests a link between T-cell memory and exhaustion, displaying exhausted markers and stem cell-like properties. Indeed, PD1^+^ TCF1^+^ CD8^+^ T-cells can expand to produce terminally differentiated PD1^+^ TCF1^−^ T-cells, while maintaining a pool of progenitor PD1^+^ TCF1^+^ T-cells [151]. This intriguing observation should be explored further, as several studies have demonstrated that TSCMs and TCMs show superior persistence and effector functions compared to effector TEMs and effector T-cells.

### 3.3. T-Cell Stemness

Effector T-cells were originally considered the optimal product for ACT due to their efficacy in killing tumor cells. However, their persistence is limited, they have poor expansion abilities, and they are easily exhausted. TSCMs are ideal candidates for ACT due to their longevity, aptitude for self-renewal, and differentiation in different T-cell populations. Several studies have demonstrated that a positive clinical response correlates with early T-cell engraftment and expansion [99,249,250,251,252,253,254,255,256,257]. CAR T-cell persistence is usually a pivotal requirement for lasting remission, despite the fact that CAR T-cells were not detectable in some patients with complete remission. Clinical studies have demonstrated that the infusion of phenotypically and functionally TSCM-like CAR T-cells (CD62L^+^, CD28^+^, and CD27^+^) results in favorable outcomes [92,99,258,259,260]. For example, CLL and multiple myeloma patients treated with anti-CD-19 CAR T-cells displayed a favorable response in association with the CD27^+^ CD45RO^−^ CD8 cell population [261,262]. Moreover, investigations in mice have demonstrated that the infusion of T-cell populations enriched in CD62L^+^ resulted in increased engraftment, expansion, and persistence, leading to lasting tumor regression [92,255,260,263,264,265]. Another example is the clinical trial NCT02348216, in which patients with refractory large B-cell lymphoma were treated with axi-cel, an anti-CD19 CAR T-cell therapy. Univariable and multivariable analyses showed that durable responses were associated with high product CCR7^+^CD45RA^+^ T cells, low baseline tumor burden, and low systemic inflammation [266]. The expression of CCR7 with the naïve cell marker CD45RA discriminates between naïve (CD45RA^+^CCR7^+^) and TCM (CD45RA^−^CCR7^+^) from TEM (CD45RA^−^CCR7^−^).

Despite this extensive body of evidence, unselected intra-tumoral or peripheral blood mononuclear cell (PBMC)-derived T-cell populations are used in most clinical trials, due to technical difficulties associated with clinical-grade manufacturing of CAR T-cells enriched in TSCMs. For instance, TSCMs represent only 2–3% of circulating human lymphocytes and require specific culturing conditions [267]. PBMC composition can vary considerably among patients due to prior chemotherapy treatments, age, previous health conditions, and history of pathogen exposure. To generate CAR T-cell products enriched in TSCMs, IL-7, IL-15, and IL-21 are used during the manufacturing process [97,268,269,270,271,272]. These cytokines promote TSCM development and expansion. Specifically, IL-21 restrains T-cell differentiation by activating STAT3 and TCF7/LEF1 transcription factors [273,274]. 4, 6-Disubstituted pyrrolopyrimidine (TWS119) is an inhibitor of the glycogen synthase kinase-3β (GSK-3β) and a robust activator of the Wnt pathway that enhances TSCM proliferation. By stabilizing β-catenin, TWS119 synergizes with IL-21 to upregulate TCF7 and LEF1 [263,275].

Vodnala et al. (2019) showed that treatment of T-cells with elevated potassium concentration in the culture medium mimics functional starvation and results in T-cells with retained stemness, self-renewal, and multipotency capacity [276]. T-cells rely on glycolysis for their growth and proliferation, especially after encountering the antigen. TEMs instead use the FAO pathway, a non-proliferative form of metabolism, and they obtain ATP mainly through OXPHOS. This was demonstrated by inhibiting the mTOR pathway and upregulating the FAO pathway [125,277]. Using selective inhibitors of Akt also leads to the generation of TEMs [278]. In response to T-cell activation, cells experience increased mitochondrial metabolism and the generation of reactive oxygen species. Studies have linked T-cell metabolism and cytotoxic efficacy, and it is likely that expansion and effector function are fueled at the cost of compromised self-renewal capacity. Therefore, reprogramming intrinsic metabolic pathways may enhance the efficacy of CAR T-cell therapy.

The T-cell “progressive differentiation model” proposed by Sallusto and Lanzavecchia (2002) suggests a hierarchical differentiation tree in which naïve T-cells, upon priming, differentiate into TEMs (Figure 3) [279]. Naïve lymphocytes are immature cells that have not been antigen-triggered. They can differentiate to form effector cells and/or memory cells. TSCM is the first stage of differentiation of memory cells, and further progression leads to the formation of TCMs. TCMs can differentiate into effector TEMs or effector cells. Further differentiation of TEMs leads to the formation of effector cells with short-life and high cytotoxic potential, although some can reconvert to TCMs. Genetic barcoding of T-cells in mice provided strong evidence for this model [280,281]. Further studies in mice, non-human primates, and humans confirmed that TSCMs are the apex of the TEM hierarchical tree [249,260,282,283,284]. TSCMs display a specific epigenetic and transcriptional profile that is very similar to TCMs, as demonstrated by genome-wide transcriptomic analysis [285,286,287]. The transcriptional network involved in the transition from precursor to mature memory cell is largely uncharacterized, however, it appears that the TCF-1 and FoxO1 transcription factors may play a pivotal role [288,289,290]. Little is known about the epigenetic program in TSCMs, however recent studies on histone methylation in different T-cell subsets revealed that chromatin accessibility is likely regulated in a progressive fashion, in accordance with the hierarchical model described above [285].

In both TSCMs and TCMs (CD62L^+^ CCR7^+^), the Wnt/β-catenin signaling pathway plays a pivotal role in cell survival. The expression of TCF1 is critical to preserve T-cell stemness, as it participates in the WNT-β-catenin signaling [291]. Generally, high levels of the TCF1 transcription factor in CD8^+^ T-cells are positively correlated with the response to checkpoint blockage treatment [150]. TSCMs express CD95 and IL-2Rβ, have higher proliferation capacity, and a greater ability to release cytokines rapidly upon activation compared to TCMs. Compared to TEMs, TSCMs and TCMs have an increased capacity for IL-2 secretion and reduced capacity for IFN-γ production, despite comparable TNF-α production and induction of telomerase to maintain replicative potential. Long-term T-cell clones mostly originate from infused TSCMs, and to a lesser degree, from the TCM subset. When TCMs differentiate into TEMs, downregulation of CCR7 and CD62L occurs, along with a reduction in migration to lymphoid tissues [249,292,293,294].

## 4. Conclusions

Despite remarkable advances in immunotherapy in the last decade, improving the efficacy of CAR T-cell therapies still faces many challenges. The limited persistence of CAR T-cells in patients is challenging and a main cause is T-cell exhaustion. Scientists have approached the issue of T-cell exhaustion with different perspectives: creating engineered T-cells that are exhaustion-resistant, modifying manufacturing conditions, or introducing new treatments such as checkpoint blockade. Further improvements to the design of CAR constructs will likely broaden their clinical applications. Promoting stemness and perfecting cell lineage composition for ACT may also represent a key approach to improve efficacy and persistence. While the results to date are still modest, the possibilities are endless, and a breakthrough may be imminent.

## Figures and Tables

**Figure 1 ijms-22-10828-f001:**
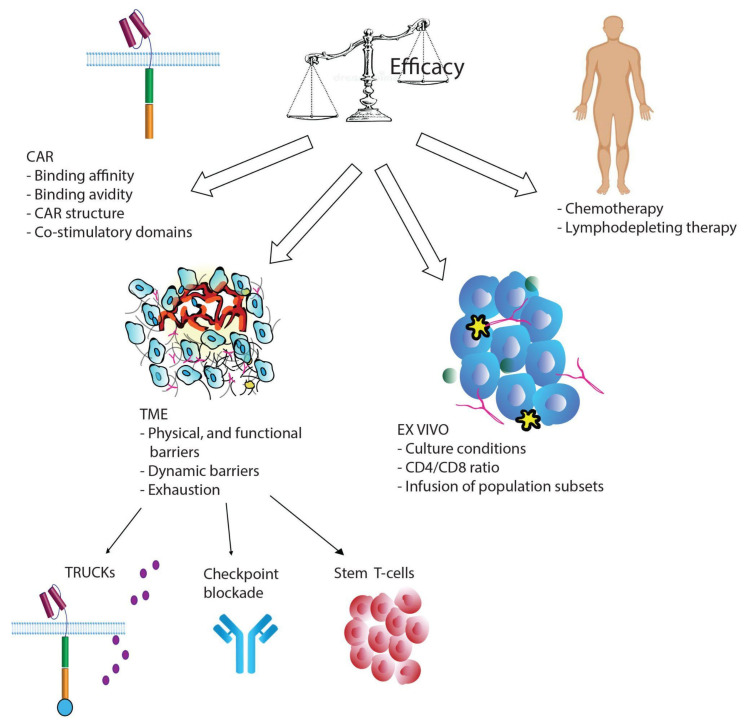
A schematic of determinants affecting the efficacy of CAR T-cell therapy and possible solutions to the issue of T-cell exhaustion.

**Figure 2 ijms-22-10828-f002:**
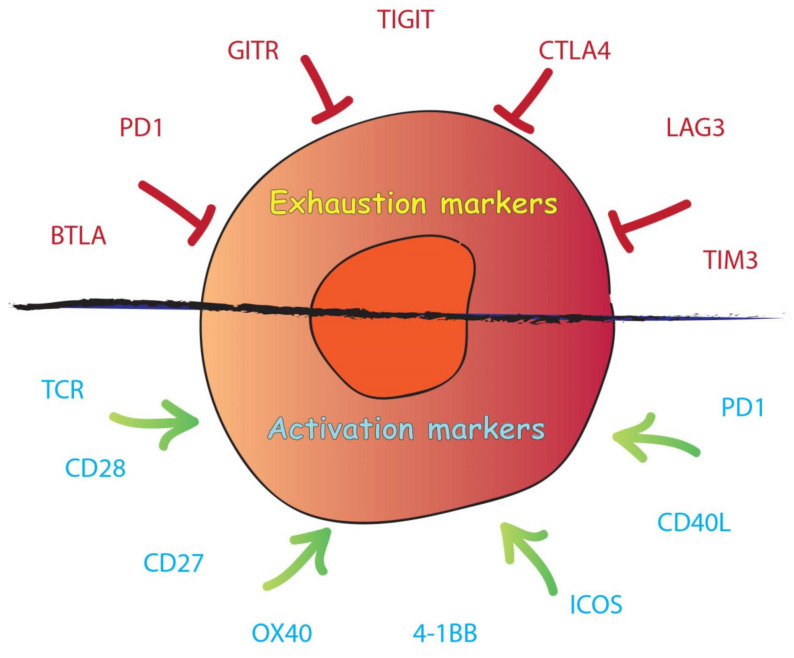
Scheme of T-cell exhaustion and T-cell activation markers.

**Figure 3 ijms-22-10828-f003:**
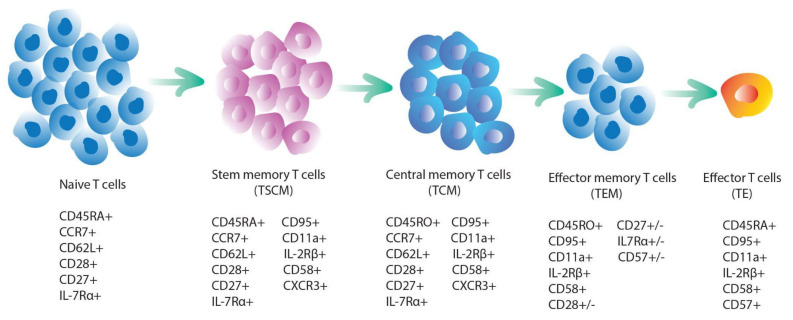
A model of T-cell differentiation and markers of T-cell subpopulations.

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
