# Peer review of "Improving CAR T-Cell Persistence"

_ijms, 2021, doi:10.3390/ijms221910828_

Round 1

Reviewer 1 Report

Reviewer comments and suggestions

Pietrobon et al reviewed in their manuscript on improving CAR T-cell therapy and discusses the merit and demerits of the treatment in cancer therapy. The study suggested the clinical benefits in solid tumors but even having the risk of relapsing due to several reasons such as poor T-cell expansion and lack of long-term persistence after adoptive transfer. For the development of effective CAR T therapies, the authors discussed several methods in the manuscript. 

The study explores several key clinical trials and highlights the molecular mechanisms determining T-cell stemness, as indorsing stemness could be the best way in improving T-cell therapies.

Below are the suggested comments to be incorporated in the revised version of the manuscript.

  1. In Introduction, first 5 lines need a reference 
  2. The manuscript needs final editing before resubmission of the manuscript because at several places the references were italic
  3. For the introduction section : authors to please do not repeat we will as seen here page number 3
  4. Section 2 The sentence need to explore here “Tolerance occurs in situations of chronic low-grade inflammation (para-inflammation) and it is dependent on the context of different cancer subtypes”
  5. T-cell exhaustion due to decreased tonic signaling and enhanced persistence please explore the reason comprehensibly.
  6. “Currently novel CAR T products are being investigated, which combine the CD28 and 4-1BB costimulatory domains”. The authors mentioned currently but provided the reference of 2019
  7. I found big sentences in the MS, Try to avoid it Other clinical trials (#NCT00586391 …… lymphoid organs 
  8. Only one study included antioxidants? Addition of antioxidants such as N-acetylcysteine to the cell culture during the manufacturing process has also been shown to inhibit effector differentiation 
  9. Is any other study reported a similar result, if yes please cite Among patients treated with high dose lymphodepletion, PFS was increased in those achieving a favorable cytokine profile as well, compared to patients that did not achieve a favorable profile  
  10. It seems that the line was not completed “Indeed, effector T-cells rely mainly on glycolytic metabolism to perform their cytotoxic functions, while the metabolism of their precursors consists of fatty acid oxidation”
  11. NR4A transcription factors, I found some time they are in capital or some time they are small
  12. “A plethora of transcription factors are” Previous para was also describing transcription factor, I think the section should be short
  13. Is this important to include “Unpublished data from Wang Ena and colleagues of whole transcriptome profiling of 200 retention samples of TILs, revealed that transcriptional pro”
  14. In LIST OF ABBREVIATIONS, two time used stem memory T cell (TSCM) stem memory T-cells (TSCM) was used
  15. References not arranged according to MDPI

Author Response

REVIEWER 1

Pietrobon et al reviewed in their manuscript on improving CAR T-cell therapy and discusses the merit and demerits of the treatment in cancer therapy. The study suggested the clinical benefits in solid tumors but even having the risk of relapsing due to several reasons such as poor T-cell expansion and lack of long-term persistence after adoptive transfer. For the development of effective CAR T therapies, the authors discussed several methods in the manuscript. 

The study explores several key clinical trials and highlights the molecular mechanisms determining T-cell stemness, as indorsing stemness could be the best way in improving T-cell therapies.

Below are the suggested comments to be incorporated in the revised version of the manuscript.

1. In Introduction, first 5 lines need a reference

Thank you for your comment. We added the reference for the first 5 lines in Introduction.

2. The manuscript needs final editing before resubmission of the manuscript because at several places the references were italic

We thank the reviewer for noticing this and have ensured references are not italicized.

3. For the introduction section : authors to please do not repeat we will as seen here page number 3

We thank the reviewer for noticing that this is repetitive and we have edited the paragraph accordingly.

4. Section 2 The sentence need to explore here “Tolerance occurs in situations of chronic low-grade inflammation (para-inflammation) and it is dependent on the context of different cancer subtypes”

Thank you for your comment. We added an additional paragraph to better explain this concept.

5. T-cell exhaustion due to decreased tonic signaling and enhanced persistence please explore the reason comprehensibly.

Thank you for your comment. We clarified the sentence and added information to explain the concept comprehensibly.

6. “Currently novel CAR T products are being investigated, which combine the CD28 and 4-1BB costimulatory domains”. The authors mentioned currently but provided the reference of 2019

We thank the reviewer for noticing this mistake and we corrected the sentence. We also added a further sentence to better explain the study previously cited.

7. I found big sentences in the MS, Try to avoid it Other clinical trials (#NCT00586391 …… lymphoid organs

We thank the reviewer for this suggestion and have split the indicated sentence into two sentences. We also edited the whole manuscript in order to split other big sentences.

8. Only one study included antioxidants? Addition of antioxidants such as N-acetylcysteine to the cell culture during the manufacturing process has also been shown to inhibit effector differentiation 

Thank you for your comment. We added more references regarding N-acetylcysteine use and its effect on T cells growth and function.

9. Is any other study reported a similar result, if yes please cite Among patients treated with high dose lymphodepletion, PFS was increased in those achieving a favorable cytokine profile as well, compared to patients that did not achieve a favorable profile  

Thank you for your comment. We checked and did not find any other study that reported direct evidence of a correlation between high dose of lymphodepletion and favorable cytokine profile.

10. It seems that the line was not completed “Indeed, effector T-cells rely mainly on glycolytic metabolism to perform their cytotoxic functions, while the metabolism of their precursors consists of fatty acid oxidation”

We are not sure what the reviewer means as this sentence is complete.

11. NR4A transcription factors, I found some time they are in capital or some time they are small

As per nomenclature guidelines, NR4A is capitalized when referring to the protein, and not capitalized (Nr4a) when referring to the gene (i.e. when referring to gene knockouts).

12. “A plethora of transcription factors are” Previous para was also describing transcription factor, I think the section should be short

Thank you for your comment. We shorten this section.

13. Is this important to include “Unpublished data from Wang Ena and colleagues of whole transcriptome profiling of 200 retention samples of TILs, revealed that transcriptional pro”

Thank you for your comment. We removed that paragraph.

14. In LIST OF ABBREVIATIONS, two time used stem memory T cell (TSCM) stem memory T-cells (TSCM) was used

We thank the reviewer for pointing this out and have removed the duplicated abbreviation.

15. References not arranged according to MDPI

We thank the reviewer for noticing this mistake and we corrected the references style according to MDPI guidelines.

Reviewer 2 Report

This is a very comprehensive and complete review on the problematic of CAR T cell persistence and it includes all the recent and extensive litterature on this topic

Comments:

Please check for spelling some minor mistakes were found e.g.

p6 stills demonstrates?

p7 in  patient's T cells

axi-cell is only put in full later in the text  and not the first time it apperars

and check throughout

Author Response

REVIEWER 2

This is a very comprehensive and complete review on the problematic of CAR T cell persistence and it includes all the recent and extensive literature on this topic

Comments:

1. Please check for spelling some minor mistakes were found e.g.

We thank the reviewer for pointing this out and have checked for spelling/grammatical errors throughout the whole manuscript.

2. p6 stills demonstrates?

We thank the reviewer for notice this and have made the correction.

3. p7 in   patient's T cells

We thank the reviewer for notice this and have made the correction.

4. axi-cell is only put in full later in the text  and not the first time it appears

We thank the reviewer for notice this and have made the correction.

5. and check throughout

Thank you for your comment. We checked throughout the manuscript for other mistakes regarding abbreviations, nomenclature, spelling and grammatical errors.

Round 2

Reviewer 1 Report

In the attached manuscript, still the references cited in the text needs to be in numbers not in et al. Please check the guidelines of MDPI Journal.